# GENERATING REALISTIC 3D MOLECULES WITH AN EQUIVARIANT CONDITIONAL LIKELIHOOD MODEL

## ABSTRACT

The number of drug-like molecules that could potentially exist is thought to be above $10^{33}$, precluding exhaustive computational or experimental screens for molecules with desirable pharmaceutical properties. Machine learning models that can propose novel molecules with specific characteristics are powerful new tools to break through the intractability of searching chemical space. Most of these models generate molecular graphs—representations that describe the topology of covalently bonded atoms in a molecule—because the bonding information in the graphs is required for many downstream applications, such as virtual screening and molecular dynamics simulation. These models, however, do not themselves generate 3D coordinates for the atoms within a molecule (which are also required for these applications), and thus they cannot easily incorporate information about 3D geometry when optimizing molecular properties. In this paper, we present GEN3D, a model that concurrently generates molecular graphs and 3D geometries, and is equivariant to rotations, translations, and atom permutations. The model extends a partially generated molecule by computing a conditional distribution over atom types, bonds, and spatial locations, and then sampling from that distribution to update the molecular graph and geometries, one atom at a time. We found that GEN3D proposes molecules that have much higher rates of chemical validity, and much better atom-distance distributions, than those generated with previous models. In addition, we validated our model's geometric accuracy by forcing it to predict geometries for benchmark molecular graph inputs, and found that it also advances the state of the art on this test. We believe that the advantages that GEN3D provides over other models will enable it to contribute substantially to structure-based drug discovery efforts.

## 1 INTRODUCTION

Identifying molecules with desirable characteristics is of fundamental importance in many fields, including drug discovery. The astronomically large number of possible drug-like compounds, however, makes an exhaustive experimental or virtual screening intractable (Polishchuk et al., 2013). As a result, a large body of work has used machine learning to explore chemical space and propose molecules with specific characteristics.

Organic molecules can be represented as graphs in which nodes are individual atoms, and edges are covalent bonds through which atoms share electrons. Each node in a molecular graph is labelled with the atomic number of its corresponding atom, and edges are labelled with the number of electrons shared in that covalent bond. Because each atom has a predetermined number of electrons with which to form bonds, only a subset of the possible edge labels result in chemically valid molecules.

The connectivity-based description of a molecule provided by molecular graphs is important for many applications, including chemical synthesis, virtual screening, and molecular dynamics simulation, and has thus motivated extensive research into generative models for molecular graphs (Gómez-Bombarelli et al., 2018; Maragakis et al., 2020; Liu et al., 2018; Shi et al., 2020; Jin et al., 2018; 2020). The functional characteristics of a given molecule, however, arise not only from its connectivity, but from its configuration in 3D space. Physical limitations on the lengths and angles of covalent bonds among specific types of atoms impose constraints on the set of geometric configurations that are compatible with a given molecular graph. Because most organic molecules

contain rotatable bonds, however, these constraints are generally not sufficient to unambiguously reconstruct the 3D coordinates of all atoms in the molecule. The probability of observing a particular 3D geometry for a given molecule is in general a function of the quantum mechanical energy of that geometry, with low-energy geometries being more likely.

Graph-based generative models are capable of producing a wide range of molecules whose atoms have the correct number and type of bonds, but their outputs do not contain any geometric information, and may include molecular graphs for which stable 3D geometries (i.e., low-energy geometries with physically realistic inter-atomic distances and bond angles) do not exist. Although it is possible that a model could learn to assess the geometric feasibility of a molecular graph without training on any geometric data, a model that is trained explicitly on 3D molecules would likely be better at generating molecular graphs with corresponding low-energy 3D geometries. The lack of geometric information is also problematic because many downstream applications require 3D information. For instance, many drug discovery efforts screen for molecules with high predicted affinity for a target protein pocket using a computational docking process to score molecule poses. It would be extremely valuable if a machine learning model could directly generate such molecules, as this would expedite the screening process. A model that is explicitly trained to recreate the geometric poses of bound molecules, rather than just molecular graphs, has the potential to produce molecules and geometries that are more conducive to binding.

To address the above shortcomings, we created GEN3D—a graph-generative model that proposes molecules with 3D coordinates. The model is rotationally and translationally equivariant, providing an inductive bias that exploits the symmetries of chemical space. GEN3D creates molecular graphs through a sequential sampling process, like other graph-based generative models (Shi et al., 2020; Liu et al., 2018), but at each step GEN3D also calculates a likelihood function over positions in 3D space, and uses that function to sample coordinates for each new atom. We show that GEN3D generates novel, chemically valid molecular graphs that have realistic low-energy geometries. In particular, we found that our model outperforms existing graph generators in its ability to create chemically valid molecular graphs, and outperforms existing 3D-generative models in the ability to create realistic geometries. In Appendix E, we also show that our pre-trained model can be tuned to generate novel molecules in geometries that score well in a virtual screening tool, which illustrates the potential application of our method in drug discovery. Finally, we demonstrate the geometric accuracy of our model by using it to sample geometries for fixed molecular graphs, and show that it achieves state-of-the-art results on the previously established benchmark of Xu et al. (2021b).

## 2 RELATED WORK

Different models construct molecular graphs using Graph Neural Networks (GNNs) in a number of different ways. CGVAE, for example, uses a GNN to create VAE latent representations for each node in a molecular graph (Liu et al., 2018). The nodes and edges of the graph are then reconstructed in a sequential decision process, guided by another GNN, in which atoms are connected one at a time to a growing molecular graph. Another model, GraphAF, also generates molecular graphs one atom at a time, but it samples molecules using a Gaussian Autoregressive Flow (Shi et al., 2020). Other models, like JT-VAE (Jin et al., 2018) and HierVAE (Jin et al., 2020), generate molecules by sequentially adding multi-atom motifs rather than individual atoms.

Because of their importance in classification tasks on point clouds and 3D molecules, there has been a great deal of interest in machine learning problems involving 3D structured data. When working with such data, it is often desirable that models be invariant or equivariant to rotations or translations of the input data, as these transformations are only an arbitrary change of coordinate systems. Approaches like SchNet (Schütt et al., 2017) and EGNN (Satorras et al., 2021b) achieve equivariance by using invariant features like pairwise distances as inputs to the model. These approaches have proven effective in many domains, but they are reflection-invariant, so they cannot distinguish between mirror-image isomers. Other approaches, like DimeNet (Klicpera et al., 2020) and SphereNet (Liu et al., 2021), achieve equivariance through message-passing schemes that encode distance and angular information, while Tensor Field Networks (Thomas et al., 2018) and SE(3)-Transformers (Fuchs et al., 2020) process data using a basis of equivariant spherical harmonic functions.

There are several prior works on generative models for 3D molecular data. Two models from the same group generate 3D molecules by outputting a voxelized grid of atomic densities, which is

then converted to a set of atoms and bonds in a secondary optimization step (Ragoza et al., 2020; Masuda et al., 2020). While efficient to implement using convolutions, the voxelized grid approach imposes a practical limit on the size of generated molecules, and is not equivariant. The process of reconstructing a molecule from a density field also suffers from relatively high rates of chemical and geometric invalidity, and the one-shot nature of the generative process does not allow for valence constraints to be enforced. Another pair of recent papers aims to generate stable 3D molecules using reinforcement learning (Simm et al., 2019a;b). These models, however, are only capable of generating molecules by placing atoms from a set corresponding to a pre-determined stoichiometry, have only been demonstrated on very small molecules, and do not generate bonding information.

Two other 3D-generative models are closely related to GEN3D: E-NF (Satorras et al., 2021a) and G-SchNet (Gebauer et al., 2019). E-NF uses an equivariant, EGNN-based normalizing flow to convert random initial atom positions into realistic molecular geometries. This model, however, has only been demonstrated on relatively small molecules, is very expensive to train, and produces molecules with low rates of chemical validity. G-SchNet is an equivariant model that autoregressively generates 3D molecules, and has recently been extended to conditionally generate molecules with desired electronic properties and molecular fingerprints (Gebauer et al., 2021). G-SchNet, however, exclusively generates atomic positions, and does not generate molecular graphs that provide the chemical bonding information required by many downstream applications. In addition, without explicitly generating bonds, one cannot use simple rules to constrain chemical properties, like atomic valences and ring sizes, during the generative process.

There are also machine learning methods that predict molecular geometries for a given molecular graph. A number of works have approached this problem by predicting inter-atomic distances from the molecular graph, and then using the predictions to generate a 3D geometry in a secondary optimization step (Xu et al., 2021a; Simm & Hernández-Lobato, 2019). Most recently, Xu et al. (2021b) proposed the ConfVAE architecture, which estimates inter-atomic distances and then optimizes molecular geometry using an end-to-end differentiable optimization procedure.

## 3 PROBABILISTIC MODEL

We will represent a molecule as a 3D-dimensional graph $G = (V, A, X)$. For a molecule with $n$ atoms, $V \in \mathbb{R}^{n \times d}$ is a list of $d$-dimensional atom features, $A \in \mathbb{R}^{n \times n \times b}$ is an adjacency matrix with $b$-dimensional edge features, and $X \in \mathbb{R}^{n \times 3}$ is a list of 3D atomic coordinates for each atom. In practice, $V$ simply encodes the atomic number of each atom, and $A$ encodes the number of shared electrons in each covalent bond. To model a chemical space of interest, we consider the distribution $p(V, A, X)$. Previous works have aimed at calculating various marginal and conditional densities with respect to this joint distribution. For example, graph-based generative models learn the marginal distribution $p(V, A) = \int_X p(V, A, X) dX$, molecular geometry prediction amounts to learning the conditional distribution $p(X|V, A)$, and 3D generative models like G-SchNet learn the distribution $p(V, X) = \int_A p(V, A, X) dA$. To learn the joint distribution $p(V, A, X)$, it is necessary to factorize the density. We choose to use the following factorization:

$$p(V, A, X) = \prod_{i=1}^{n} p(V_{:i}, X_{:i}, A_{:i}|V_{:i-1}, A_{:i-1}, X_{:i-1}) \cdot p(\text{stop}|V, A, X)$$

$$= \prod_{i=1}^{n} p(X_{:i}|V_{:i}, A_{:i}, X_{:i-1}) p(A_{:i}|V_{:i}, A_{:i-1}, X_{:i-1}) p(V_{:i}|V_{:i-1}, A_{:i-1}, X_{:i-1}) \cdot p(\text{stop}|V, A, X)$$

Here, $n$ is the number of atoms in the input graph, and $V_{:i}$, $A_{:i}$ and $X_{:i}$ indicate the graph $(V, A, X)$ restricted to the first $i$ atoms. Computing $p(V_{:i}|V_{:i-1}, A_{:i-1}, X_{:i-1})$ is relatively simple because it amounts to predicting a single atom type based on a 3D graph $(V_{:i-1}, A_{:i-1}, X_{:i-1})$. Calculating $p(A_{:i}|V_{:i}, A_{:i-1}, X_{:i-1})$ is more complex because it involves a prediction over a new row of the adjacency matrix. More concretely, computing the conditional density of $A_{:i} \in \mathbb{R}^{i \times i \times b}$ amounts to computing a joint density over the new entries of the adjacency matrix $A_{i,1}, \ldots, A_{i,i-1} \in \mathbb{R}^b$. To solve this problem, we further decompose this distribution:

$$p(A_{:i}|V_{:i}, A_{:i-1}, X_{:i-1}) = p(A_{i,1}, \ldots, A_{i,i-1}|V_{:i}, A_{:i-1}, X_{:i-1}) = \prod_{j=1}^{i-1} p(A_{i,j}|A_{i,:j-1}, V_{:i}, A_{:i-1}, X_{:i-1})$$

Intuitively, $A_{i,1}, \ldots, A_{i,i-1}$ represent the edges from atom $i$ to atoms $1 \ldots i-1$.

Finally, estimating the density $p(X_{:i}|V_{:i}, A_{:i}, X_{:i-1})$ involves modeling a continuous distribution over positions $X_i \in \mathbb{R}^3$ for atom $i$. To accomplish this, we assume $X_i$ belongs to a finite set of points $\mathcal{X}$, and model its probability mass as a product of distributions over angles and distances:

$$p(X_i|V_{:i}, A_{:i}, X_{:i-1}) = \frac{1}{C} \prod_{j=1}^{i-1} p(||X_i - X_j|| \mid V_{:i}, A_{:i}, X_{:i-1})$$

$$\cdot \prod_{(j,k) \in I} p(\text{Angle}(X_i - X_k, X_j - X_k) \mid V_{:i}, A_{:i}, X_{:i-1})$$

Intuitively, $p(||X_i - X_j|| \mid V_{:i}, A_{:i}, X_{:i-1})$ predicts the distances from each existing atom to the new atom, and $p(\text{Angle}(X_i - X_k, X_j - X_k) \mid V_{:i}, A_{:i}, X_{:i-1})$ predicts the bond angles of connected triplets of atoms involving atom $i$. $I$ is a set of pairs $(j, k)$ where atom $k$ is connected to atom $i$, and atom $j$ is connected to atom $k$. "Angle" denotes the angle between two vectors. $C$ is a normalizing constant derived from summing this density over all of $\mathcal{X}$. To increase the computational tractability of estimating this factorized density, we assume that the nodes in the molecular graph $(V, A, X)$ are listed in the order of a breadth-first traversal over the molecular graph.

## 4 ARCHITECTURE AND TRAINING

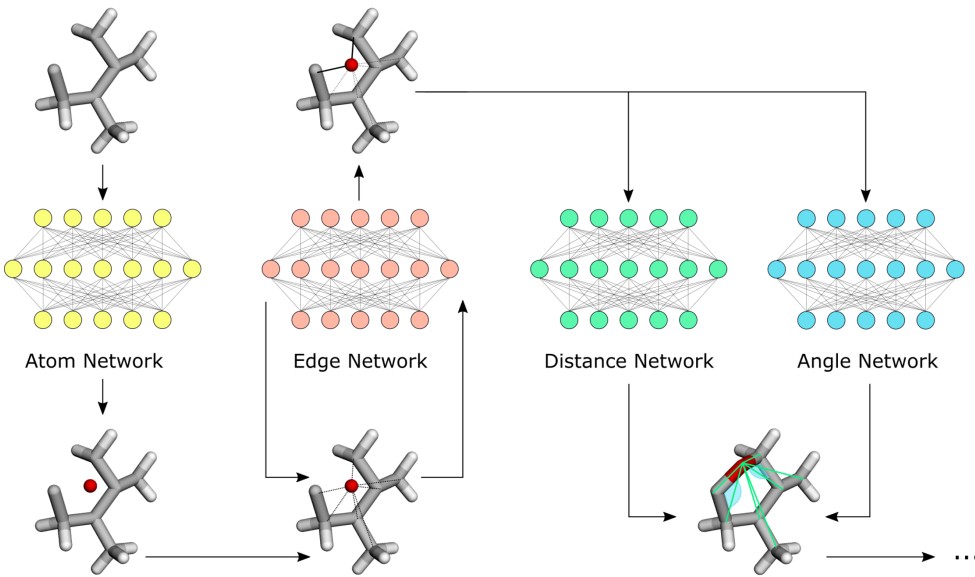

Figure 1: The process by which GEN3D samples a 3D molecule.

The GEN3D architecture uses a collection of four equivariant neural networks: the atom network (denoted $F_A$), the edge network (denoted $F_E$), the distance network (denoted $F_D$), and the angle network (denoted $F_\theta$). Each of these networks is implemented as a 7-layer EGNN with a hidden dimension of 128, as described in Satorras et al. (2021b). We found that adding batch normalization to the atom, distance, and angle networks improved stability (Ioffe & Szegedy, 2015). An EGNN network takes in a 3D graph as input, and outputs vector embeddings for each node in the input graph. GEN3D also uses four simple MLPs, $D_A, D_E, D_D$, and $D_\theta$, to decode the output embeddings of each EGNN into softmax probabilities. GEN3D's subnetworks are used to compute the components of the factorized density above as follows:

$$p(V_{:i}|V_{:i-1}, A_{:i-1}, X_{:i-1}) = \text{Softmax}(D_A(\text{SumPool}(F_A(V_{:i-1}, A_{:i-1}, X_{:i-1}))))$$

$$p(A_{i,j}|A_{i,:j-1}, V_{:i}, A_{:i-1}, X_{:i-1}) = \text{Softmax}(D_E(F_E(A_{i,:j-1}, V_{:i}, A_{:i-1}, X_{:i-1})_j))$$

$$p(||X_i - X_j|| \mid V_{:i}, A_{:i}, X_{:i-1}) = \text{Softmax}(D_D(F_D(V_{:i}, A_{:i}, X_{:i-1})_j))$$
$$h = F_\theta(V_{:i}, A_{:i}, X_{:i-1})$$
$$p(\text{Angle}(X_i - X_k, X_j - X_k) \mid V_{:i}, A_{:i}, X_{:i-1}) = \text{Softmax}(D_\theta(h_j, h_k))$$

Note that the predicted distance and angle distributions are discrete softmax probabilities. These discrete distributions correspond to predictions over equal-width distance and angle bins. Because all of the EGNN-computed densities are insensitive to translations and rotations of the input graph, the full product density is also insensitive to these transformations.

At training time, we compute a breadth-first decomposition of a graph $(V, A, X)$. The subnetworks of GEN3D are trained to autoregressively predict the next atom types, edges, distances, and angles in this decomposition according to the model described above. We use cross-entropy losses to penalize the model for making predictions that deviate from the actual next tokens in the breadth-first decomposition. While the model's density is not invariant across different breath-first decompositions of the same molecule, we resample each molecule's decomposition at every epoch, so the model should learn to ascribe equal densities to different rollouts of the same molecule.

The training algorithm is provided in detail in Appendix F. Our experiments use the Adam optimizer with a base learning rate of 0.001 (Kingma & Ba, 2014). All models were able to train in approximately one day on a single NVIDIA A100 GPU. The model is trained using teacher forcing, so it only learns to make accurate predictions when given well-formed structures as autoregressive inputs. This could make the model more brittle at generation time because it will struggle to recover from its own mistakes in previous iterations. To increase geometric robustness and mitigate this issue, we add uniform random noise of up to .05 Å to the atomic coordinates during training for all datasets.

To sample a 3D molecule from a trained GEN3D model, we can start with a single initial atom or a larger molecular fragment. First, the atom network computes a discrete distribution over new atom types to add, from which a new atom type can be sampled multinomially. The edge network is then used to sequentially sample the edge types joining the new atom to each of the previously generated atoms. The distance and angle networks compute distributions over interatomic distances and bond angles involving the newly sampled atom. To sample the new atom's position, we construct the discrete set of points $\mathcal{X}$ as a fine grid surrounding the previously-generated atoms, and assign each point a probability according to the model's distance and angle predictions. Finally, the new atom's position is sampled multinomially from the set $\mathcal{X}$. The resulting molecular graph, which has been extended by one atom, is then fed back into the autoregressive sampling procedure until a stop token is generated. This sampling process is illustrated in Figure 1, and described in detail in Appendix H.

## 5 RESULTS

We trained GEN3D to generate 3D molecules from three datasets: QM9, GEOM-QM9 and GEOM-Drugs (Ramakrishnan et al., 2014; Axelrod & Gómez-Bombarelli, 2020). QM9 contains 134,000 small molecules with up to nine heavy atoms (i.e., not including hydrogen) of the chemical elements C, N, O, and F. Each molecule has a single set of 3D coordinates obtained via Density Functional Theory calculations, which approximately compute the quantum mechanical energy of a set of atoms in 3D space. GEOM-QM9 contains the same set of compounds as QM9, but with multiple geometries for each molecule. GEOM-Drugs also has multiple geometries for each molecule, and contains over 300,000 drug-like compounds with more heavy atoms and atomic species than QM9.

On QM9, we trained one version of the model with heavy atoms only, and one version with hydrogens. To ensure the quality of our geometric data, we used OpenBabel (O'Boyle et al., 2011) to convert the coordinates from the QM9 source files into SDF files, which contain both coordinates and connectivity information inferred from inter-atomic distances. We discarded all molecules for which the inferred connectivity did not match the intended SMILES string (Weininger, 1988) from the QM9 source data, leaving approximately 124,000 molecules with SDF-formatted bonding information. We used 100,000 of these molecules for training, with the remaining molecules used for validation. For GEOM-QM9 we trained on 200,000 molecule-geometry pairs, and excluded all SMILES strings from the test set of Xu et al. (2021a). For GEOM-Drugs we trained with heavy atoms only, using 50,000 randomly chosen molecule-geometry pairs for training. After 60 epochs of training, GEN3D was able to generate highly realistic 3D molecules from all of these datasets. Visualizations of GEN3D samples from QM9 and GEOM-Drugs are shown in Figure 2.

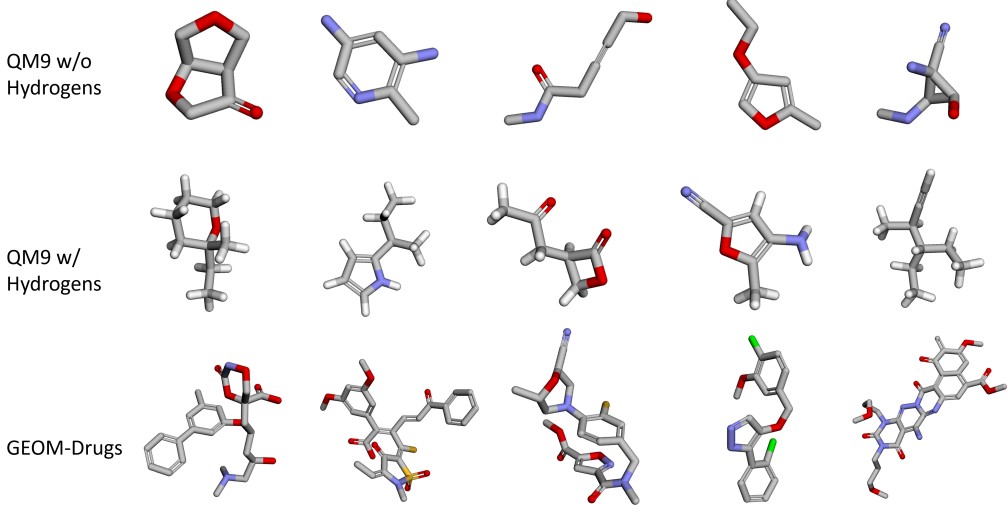

Figure 2: Samples of molecules produced by GEN3D when trained on various training sets listed in the first column. These examples have been curated to showcase the qualitative diversity of molecules created by GEN3D. For a larger number of raw samples, see Appendix A.

## 5.1 CHARACTERISTICS OF GENERATED MOLECULAR GRAPHS

To assess the quality of our generated molecules, we began by analyzing the characteristics of our generated molecular graphs with QM9. We assessed the percentages of novel and unique molecular graphs generated by the heavy atom QM9 model in a sample of 10,000 molecules. A novel molecular graph is defined as a graph not present in the training data. The uniqueness rate is defined by the number of distinct molecular graphs generated, divided by the total number of molecules generated. We focused on the heavy atom model for these metrics because most existing molecular graph generators use only heavy atoms. As mentioned previously, we can mask-out atom and edge selections at each step of the generative process such that 100% of generated molecular graphs have the correct number of bonds for each atom. We can also assess the percentage of molecules that are chemically valid, even when no masking is performed, and doing so gives an indication of the model's understanding of fundamental chemical constraints. Using the results for novelty, validity, and uniqueness metrics, we compared our approach against GraphAF and CGVAE, which are two recent molecular graph generators that also add one atom at a time. We also compared against GEN3D−, a geometry-unaware baseline that was created by removing the geometric networks from GEN3D and setting all positional inputs to 0. These results are reported in Table 1.

Table 1: Properties of QM9 Molecular Graphs (Heavy Atoms)

| Model | Validity | Validity (w/o check) | Uniqueness | Novelty | Strain |
|---|---|---|---|---|---|
| CGVAE | 100% | 49.55% | 97.09% | 88.18% | 70.84% |
| GraphAF | 100% | 67% | 94.51% | 88.93 % | - |
| GEN3D− | 100% | **99.79%** | 95.59% | 25.93% | 90.97% |
| GEN3D (ours) | 100% | **98.80%** | 94.33% | 33.18% | **93.36%** |
| QM9 (truth) | 100% | 100% | - | - | 92.41% |

We found that, even without imposing checks at generation time, GEN3D produces molecules that obey valence constraints 98.8% of the time after training on QM9. This far exceeds the unchecked validity rate of 67% achieved by GraphAF, suggesting that GEN3D has a better understanding of the basic rules of chemistry. Interestingly, our geometry-free baseline achieves 99.8% validity, suggesting that our improvements in chemical validity come from architectural differences that are unrelated to the generation of 3D geometries. GEN3D achieves a uniqueness rate of 94.3%, which is simi-

lar to the rates for GraphAF and CGVAE. We also assessed the geometric feasibility of generated graphs by converting them into 3D coordinates using CORINA (Sadowski & Gasteiger, 1993), and then computing the volume of the tetrahedron enclosed by each $sp^3$ tetrahedral center, with vertices located 1 Å along each tetrahedral bond. Graphs that could not be converted with CORINA, or contained tetrahedral centers with volumes less than 0.345 cubic Å, were classified as being overly strained. This test was adapted from Ruddigkeit et al. (2012), where it was used to filter overly strained graphs from the GDB-17 dataset. We found that GEN3D produced fewer overly strained molecules than other models, including GEN3D−, suggesting that explicitly generating molecular geometries helps bias the model towards stable compounds.

Interestingly, GEN3D showed a novelty score of only 33.18%, which is far lower than the scores reported for previously published models. We are confident, however, that our model is not overfit to its training data. Of molecules created by GEN3D that matched a molecule in our QM9 dataset, 18.6% matched a molecule in the held-out validation set. Based on the relative sizes of our validation and training sets, an unbiased model would generate molecules matching the validation set 19.5% of the time. Our model thus suffers from minimal overfitting, despite its lower novelty score.

When considering the composition of the QM9 dataset, it is not particularly surprising that GEN3D achieved a relatively low novelty score. QM9 contains all physically plausible compounds with up to nine heavy atoms of C, O, N, and F (Ramakrishnan et al., 2014). It is derived from the massive GDB-17 dataset, which enumerates the physically plausible compounds with up to 17 heavy atoms (Ruddigkeit et al., 2012). For a model to generate a molecular graph outside of the QM9 dataset, that molecule must thus have more than nine heavy atoms, or violate a basic chemical constraint that is satisfied by all compounds in the GDB-17 database. Models like CGVAE and GraphAF report >90% novelty rates when trained on the majority of compounds in QM9, meaning that their generated molecules violate the constraints of the QM9 data distribution very frequently. To further explain the discrepancy in novelty scores between GEN3D and other models, we analyzed why molecules generated by CGVAE and GEN3D are present or absent from QM9, and present the results in Appendix B. We found that CGVAE produces far more molecules with over nine heavy atoms, which artificially increases its novelty score. In contrast, GEN3D almost always generates molecules with 9 or fewer heavy atoms. This is not because of any hard cap on the number of atoms in molecules produced by GEN3D. Rather, GEN3D has learned to obey the limit of nine heavy atoms found in the training data, whereas CGVAE has not. Because GEN3D learns to stay within this limit so accurately, and QM9 is an exhaustive dataset, this leads to lower novelty scores that approach the proportion of the dataset that was withheld during training—the optimal outcome for a model that aims to generate the region of chemical space enumerated by QM9.

## 5.2 ACCURACY OF MOLECULAR GEOMETRIES

After assessing the quality of our generated graphs, we sought to verify the quality of the 3D geometries produced by GEN3D. We compared our model to E-NF, a non-equivariant version of E-NF called GNF-attention, and G-SchNet, which are the only other published models that generate samples from the distribution of 3D QM9 molecules. Both E-NF and G-SchNet produce the positions of heavy atoms and hydrogens as the output of their generative process. We compared these models to our all-atom QM9 model. The E-NF paper reports atomic stability as the percentage of atoms that have a correct number of bonds, and molecular stability as the fraction of all molecules with the correct number of bonds for every atom. We report these metrics in Table 4, and compare GEN3D to E-NF, G-SchNet, and related baselines.

GEN3D outperformed all other models, achieving 97.5% molecular stability without any valence masking, compared to 77% for G-SchNet and 4.3% for E-NF. In order to assess the geometric realism of the generated molecules, the authors of E-NF computed the Jensen-Shannon divergence between a normalized histogram of inter-atomic distances and the true distribution of pairwise distances from the QM9 dataset. We also computed this metric, and found that GEN3D advances the state of the art, reducing the JS divergence by a factor of two over G-SchNet and a factor of four over E-NF. The fact that GEN3D substantially outperforms E-NF and G-SchNet, both of which only generate coordinates and do no generate bonding information, suggests that generating bonds as well as coordinates significantly increases the quality of generated molecules.

To confirm this, we conducted a systematic ablation study (see Appendix G) in which we successively removed the angle and edge networks of GEN3D to produce a baseline model that is very similar to G-SchNet. We found that performance in both geometric and chemical accuracy metrics dropped continuously as we removed these features, and that our baseline model performed very similarly to G-SchNet. In addition, GEN3D is much cheaper to train than E-NF's flow-based generative process, and it is applicable to larger, drug-like molecules. These comparisons are reported in Table 2, and the true and learned histograms of pairwise distances are plotted in Appendix C. In order to be consistent with the E-NF paper, the Jensen-Shannon divergence was only computed between generated and QM9 molecules with exactly 19 total atoms.

Table 2: Properties of QM9 Molecules (3D models with hydrogens)

| Model | Atom Stability | Mol Stability | Distance JS |
|---|---|---|---|
| GNF-attention | 72% | .3% | .007 |
| E-NF | 84% | 4.2% | .006 |
| G-SchNet | 98.7% | 77% | .0031 |
| GEN3D (w/o check) | **99.7%** | **97.5%** | **.0014** |
| GEN3D (w/ check) | **99.87%** | **99.1%** | **.0014** |
| QM9 (truth) | 99.99% | 99.9% | 0 |

The Jensen-Shannon divergence metric provides confidence that GEN3D is, on average, generating accurate molecular geometries. This metric, however, is relatively insensitive to the correctness of individual molecular geometries because it only compares the aggregate distributions of distances. In order to further validate the accuracy of GEN3D's generated geometries, we used GEN3D to predict the geometries of specific molecular graphs, and compared its accuracy with purpose-built tools designed for molecular geometry prediction, such as the model described in Xu et al. (2021b). This evaluation amounts to verifying the accuracy of the conditional distribution $p(X|V, A)$ when the joint distribution $p(V, A, X)$ is learned by GEN3D. We approximate this conditional distribution by using a search algorithm to identify geometries $X$ that give a high value to $p(V, A, X)$, as calculated by GEN3D when $V$ and $A$ are known inputs. This method is described in detail in Appendix D.

To evaluate the ability of GEN3D to predict molecular geometries, we trained GEN3D to generate molecules from GEOM-QM9 (Axelrod & Gómez-Bombarelli, 2020). We then followed the evaluation protocol described in Xu et al. (2021a) with the same set of 150 molecular graphs, which were excluded from our training set. As in prior works, we predicted an ensemble of geometries, and then computed COV and MAT scores with respect to the test set. The COV score measures what fraction of reference geometries have a "close" neighbor in the set of generated geometries, where closeness is measured with an aligned RMSD threshold; we used a threshold of 0.5 Å, following Xu et al. (2021a). The MAT score summarizes the aligned RMSD of each reference geometry to its closest neighbor in the set of generated geometries (for the full evaluation protocol, see Xu et al. (2021a)).

GEN3D achieved results that are among the best for published models on both metrics. In particular, its MAT scores outperformed all prior methods that do not refine geometries using a rules-based force field. We compared GEN3D with previous machine learning models for molecular geometry prediction, as well as the ETKDG algorithm implemented in RDKit (which predicts molecular geometries using a database of preferred torsional angles and bond lengths (Riniker & Landrum, 2015)). Table 3 shows the results of these comparisons, and Figure 3 visualizes representative geometry predictions. These data indicate that GEN3D is accurately sampling from the joint distribution of molecular graphs and molecular geometries.

## 6 CONCLUSIONS

In this paper, we have introduced GEN3D: an autoregressive model for generating molecules in 3D space that is rotationally and translationally equivariant. We have demonstrated that GEN3D produces 3D molecules that are chemically stable and geometrically realistic, and achieves state-of-the-art results on multiple benchmarks in molecular machine learning.

We believe models like GEN3D will have many practical applications in chemistry and drug discovery. A natural extension of GEN3D is a model that generates molecules in a 3D space occupied by

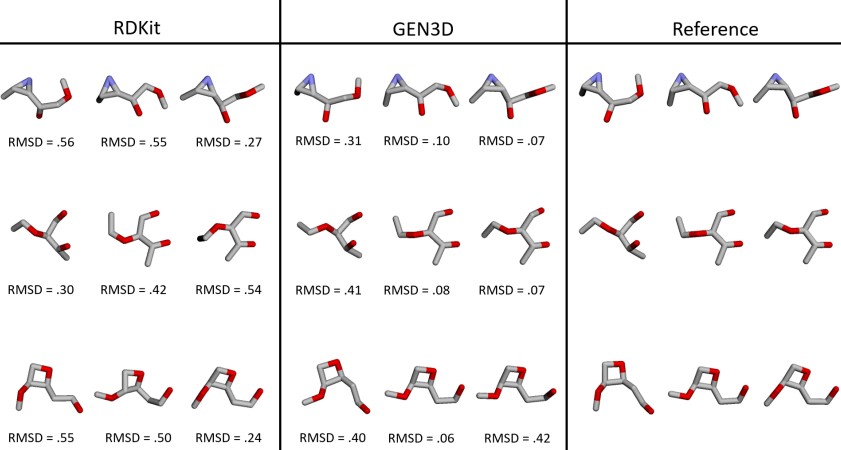

Figure 3: Predicted geometries for GEOM-QM9. The right columns contains reference geometries, and the left two columns show the nearest neighbor to the reference geometries among the geometries generated by RDKit and GEN3D.

Table 3: Geometry Prediction on GEOM-QM9

| Metric | COV (%) | | MAT (Å) | |
|---|---|---|---|---|
| | Mean | Median | Mean | Median |
| GraphDG | 55.09 % | 56.47 % | 0.4649 | 0.4298 |
| CGCF | 69.60 % | 70.64 % | 0.3915 | 0.3986 |
| ConfVAE | 77.98 % | 82.82 % | 0.3778 | 0.3770 |
| RDKit | 80.68 % | 87.50 % | 0.3349 | 0.3245 |
| GEN3D (ours) | 73.62 % | 77.14 % | **0.3168** | **0.3049** |

features of a protein pocket, such as pharmacophore features or amino acid side chain atoms. Such a model could learn to generate potential binding molecules directly inside a protein pocket. The generated molecules and geometries could then move directly into downstream stability screenings using molecular dynamics simulations or docking programs. This approach could greatly decrease the amount of computation needed to select and verify promising drug molecules. Quantum mechanical energy calculations, docking scores, and molecular dynamics simulation results from generated poses could also be used to train the model via reinforcement learning.

Conditional likelihood models like GEN3D can also work together with other drug discovery methods. For instance, if a virtual screening campaign identifies several small molecules that each bind tightly to a different region in a protein pocket, GEN3D could be used to propose large molecules that retain the respective poses of the small molecules, and join them together in a chemically and geometrically feasible way. Alternatively, if a screening campaign identifies a family of small molecules that each bind to the same region of a protein pocket and share a moiety in the context of their bound poses, GEN3D could also be used to extend this common molecular scaffold in new ways. Such applications of GEN3D could help with the enumeration of possible side chain substitutions and the closure of rings, which are common tasks is drug design.

While GEN3D already demonstrates impressive performance, it only stands to benefit from recent architectural developments in 3D machine learning. For example, GEN3D is currently reflection-invariant due its use of EGNN subunits, but this could be fixed by substituting the EGNN with reflection-sensitive networks like DimeNet or SphereNet (Klicpera et al., 2020; Liu et al., 2021).

Beyond the fields of chemistry and drug discovery, GEN3D provides a simple equivariant approach for generating 3D graphs, and is much cheaper to train than the flow-based generative models used in previous works. We hope GEN3D and similar subsequent models will find diverse uses in generating and optimizing structures in 3D space.

## REPRODUCIBILITY STATEMENT

In order to facilitate reproducibility, we plan to release the source code for the GEN3D model prior to publication, together with pre-trained GEN3D model weights and our curated version of the QM9 dataset.

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

# Appendices

## A    ADDITIONAL SAMPLE MOLECULES

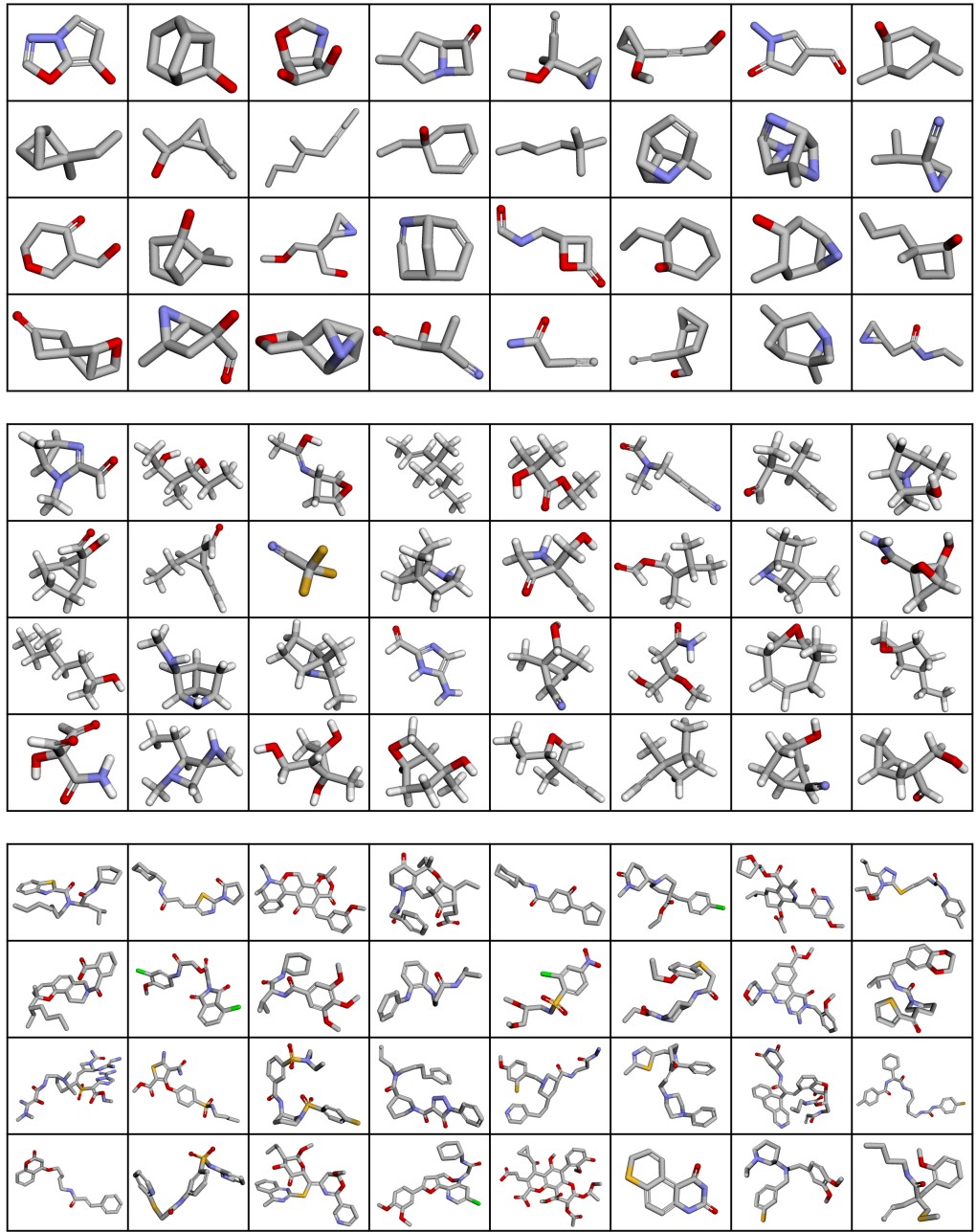

Figure S1: Unfiltered, independent random samples from GEN3D models trained on QM9 without hydrogens (top), QM9 with hydrogens (middle), and GEOM-Drugs (bottom).

## B   COMPOSITION OF GENERATED MOLECULES

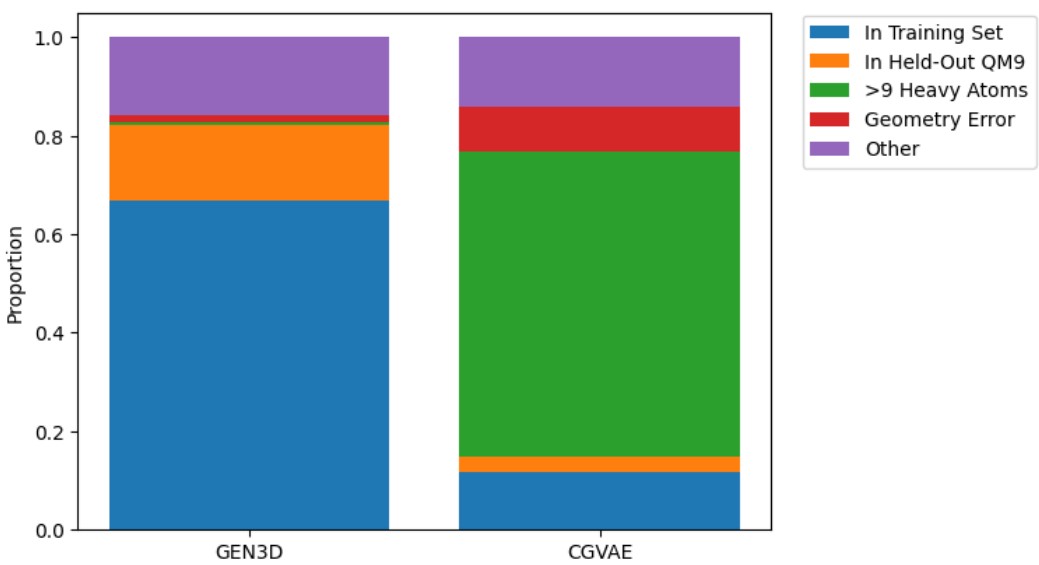

Figure S2: The composition of molecules generated from GEN3D and CGVAE.

## C    DISTRIBUTION OF INTER-ATOMIC DISTANCES

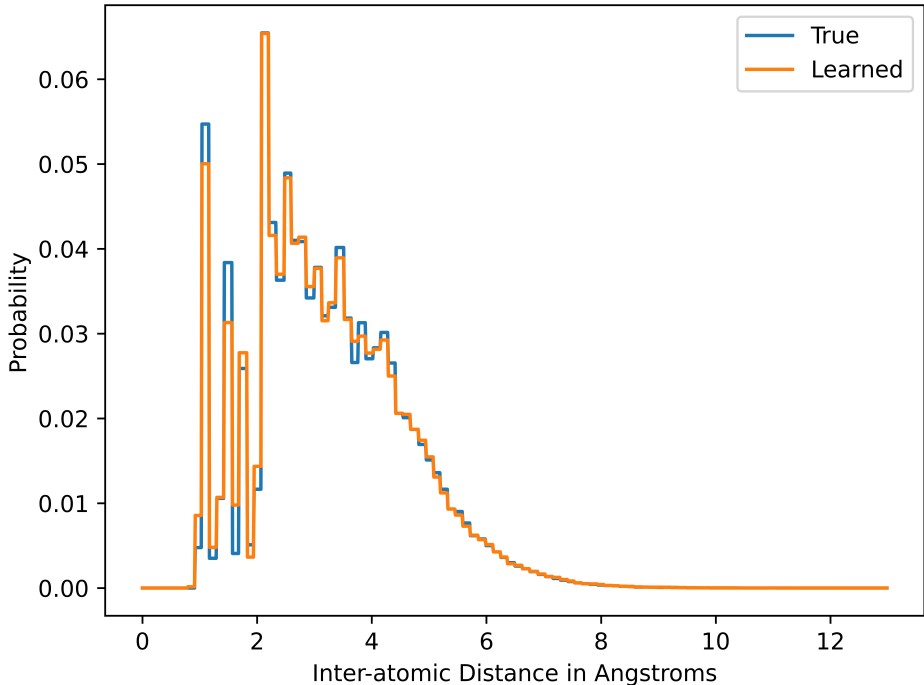

Figure S3: Histograms of inter-atom distances for GEN3D-generated and QM9 molecules with 19 total atoms.

# D  MOLECULAR GEOMETRY PREDICTION

In order to predict the geometry of a specific molecular graph, we utilized Dijkstra's algorithm to search for geometries of those molecules that are assigned a high likelihood (Dijkstra, 1959). The given molecular graph is unrolled in a breadth-first order, so predicting the molecule's geometry amounts to determining a sequence of positions for each atom during the rollout. If atomic positions are discretized, then the space of possible molecular geometries forms a tree. Each edge in the tree can be assigned a likelihood by GEN3D. Predicting a plausible geometry thus amounts to finding a path where the sum of the log-likelihoods of the edges is large. This can be accomplished using standard graph search algorithms like A* or Dijkstra's algorithm. The geometry prediction algorithm is presented in Algorithm 1 below.

---

**Algorithm 1** Molecular Geometry Search using GEN3D

---

**Input:** A molecular graph $G = (V, A)$ with $n$ atoms and a number of iterations $T$.
Assume $G$ has been processed such that the nodes of $V$ and $A$ and in breadth-first order.
$Q \leftarrow EmptyPriorityQueue()$
$Q.push((0, 1, [0.0, 0.0, 0.0]))$         ▷ Queue entries have the form (Negative Log Likelihood,
                                                    Number of Atoms, Position Matrix)

$t \leftarrow 0$
$results \leftarrow []$
**while** $t < T$ **do**
    $nll, i, X \leftarrow Q.dequeue()$
    **if** $i == n$ **then**
        $results.append((nll, X))$
    **else**
        **for** $j$ **in** $\{1..i\}$ **do**        ▷ Find an atom that is a neighbor of the next atom in the rollout
            **if** $argmax(A_{i+1,j}) > 0$ **then**
                $neigh \leftarrow j$
            **end if**
        **end for**
        $points \leftarrow MakeGrid(X[neigh])$    ▷ Create a fine grid of potential positions around the
                                              neighbor atom
        **for** $x$ **in** $points$ **do**
            $dst\_lik = p_D(x|V_{:i+1}, A_{:i+1}, X)$
            $ang\_lik = p_\theta(x|V_{:i+1}, A_{:i+1}, X_{:i})$
            $atm\_lik = p_A(V_{:i+2}|V_{:i+1}, A_{:i+1}, \text{StackCols}(X, x))$
            $edg\_lik = p_E(A_{:i+2}|V_{:i+1}, A_{:i+1}, \text{StackCols}(X, x))$
            $nll \leftarrow nll - \log(dst\_lik) - \log(ang\_lik) - \log(atm\_lik) - \log(edg\_lik)$
            $Q.push((nll, i + 1, \text{StackCols}(X, x)))$
        **end for**
    **end if**
**end while**
**return** $results$

---

We found this procedure to be effective and computational feasible for molecules in GEOM-QM9. The tree of potential geometries expands rapidly with increased numbers of atoms, however, limiting its efficiency for large molecules. Future work could identify search heuristics that make this approach more efficient.

## E  DEMONSTRATION OF PROPERTY OPTIMIZATION

We evaluated the ability of GEN3D to generate 3D molecules in poses that have favorable predicted interactions with a target protein pocket, as evaluated by the ROCS virtual screening algorithm (Grant et al., 1996). We started from a GEN3D model trained on GEOM-drugs (we call this model GEN3D-gd). We curated a large pre-existing library of 62.9-million compounds, containing up to 250 molecular geometries generated with OpenEye Omega (Perola & Charifson, 2004), for each compound, and screened the resulting 13.8-billion conformations against our target pocket using ROCS. We then selected the top 1000 scoring geometries belonging to distinct molecular graphs from the library, and we fine-tuned GEN3D-gd on these 1000 3D molecules for 100 epochs (we call the resulting model GEN3D-ft).

To evaluate the ability of GEN3D to learn chemical and geometric features that are conducive to binding the pocket, we generated 10,000 molecules with 3D coordinates from GEN3D-gd and GEN3D-ft. In addition, we took the molecular graphs generated by GEN3D-ft and recalculated molecular geometries for them using OpenEye Omega. We excluded molecules generated by GEN3D-ft if the molecular graph overlapped with the fine-tuning set (2.07% of the total), and scored the remainder using ROCS. We found that fine-tuning significantly increased the scores of generated compounds. Because GEN3D-ft was fine-tuned on high-scoring molecular geometries, the molecular geometries it generated implicitly include information about the target geometry that were unavailable to GEN3D-gd and OpenEye Omega. As a result, the scores for GEN3D-ft geometries were, on average, better than those generated by other methods. These results are shown in Figures S4.

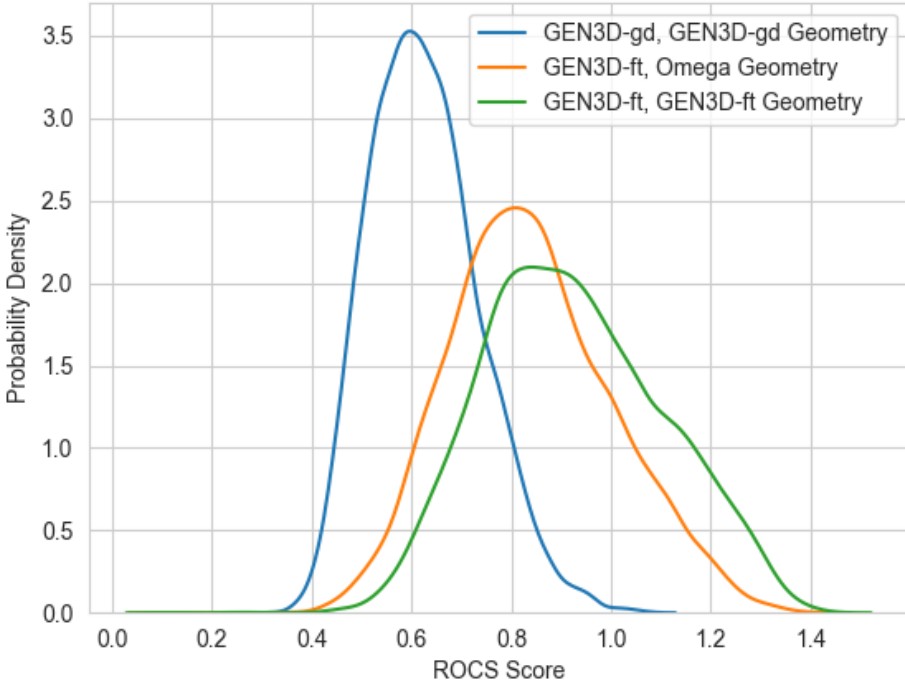

Figure S4: The probability densities of ROCS scores for molecular graphs and geometries generated by GEN3D-gd (blue), molecular graphs generated by GEN3D-ft with the Omega geometries (orange), and molecular graphs and geometries generated by GEN3D-ft (green). Higher ROCS scores indicate that a model is better able to produce desirable molecules.

Ideally, this training procedure would allow our models to generate strong binders that are significantly different from those in the fine-tuning set. To compare each model's ability to produce both high-quality and novel compounds, we picked the top 2% of molecules (by ROCS score) generated

by each model, and plotted their ROCS scores against their maximum Tanimoto similarity coefficient to an element of the set used for fine-tuning. A Tanimoto similarity coefficient (also called a Jaccard coefficient of community) ranges from fully dissimilar at 0.0 to identical at 1.0, and can be used as a measure of the structural closeness of two molecular graphs (Rogers & Tanimoto, 1960; Jaccard, 1912). It is computed by representing two molecules with Extended-Connectivity Fingerprints, which are essentially lists of activated bits corresponding to substructures present in each molecule (Rogers & Hahn, 2010). (Here, we used RDKit's implementation of Morgan fingerprints with 2048 bit, radius 2, and without chirality (Landrum et al., 2021).)

We found that GEN3D-ft generated molecules with high ROCS scores across a wide range of Tanimoto similarities to the fine-tuning set. Molecules generated by GEN3D-ft had significantly higher scores than those generated by GEN3D-gd, even when comparing molecules from each model with comparable similarities to the fine-tuning set. In this particular instance, the highest ROCS scoring molecule generated by GEN3D-ft had a Tanimoto similarity to the fine-tuning set of about 0.4. These results are shown in Figure S5.

These experiments indicate that GEN3D is able to shift its generative distribution into specific regions of chemical and geometric space.

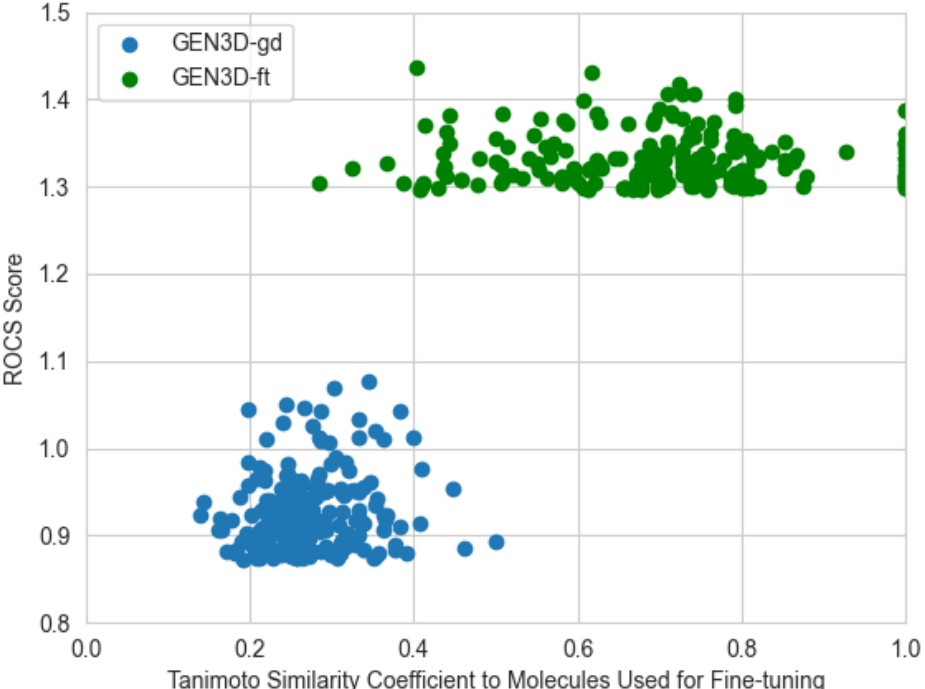

Figure S5: The similarity to the fine-tuning dataset of the molecules with the top 2% of ROCS scores. For clarity, we include those molecules that exactly match a member of the fine-tuning training set (i.e., they have a Tanimoto similarity coefficient of 1.0).

# F DETAILED TRAINING PROCEDURE

In order to train GEN3D, we decompose a 3D molecule into a sequence of partially completed molecular graphs, to which new atoms, edges, and spatial locations are added sequentially. We use GEN3D to autoregressively predict the next atom types, edge types, interatomic distances, and bond angles in this sequence of partially completed graphs, and penalize incorrect predictions using a cross-entropy loss. Algorithm 2 describes this process in detail.

---

**Algorithm 2** Training GEN3D

---

**Initial:** Parameters $\phi$ of GEN3D subnetworks $F_A, D_A, F_E, D_E, F_D, D_D, F_\theta, D_\theta$.
**while** $\phi$ not converged **do**
    Sample a 3D molecule from the training set. Let $n$ denote the number of atoms.
    Perform a Breadth-First search, producing a graph $G = (V, A, X)$ with the nodes.
    in breadth-first order.
    Let $G_{:i} = (V_{:i}, X_{:i}, A_{:i})$ denote the partially complete graph restricted to the atoms
    encountered by step $i$ of the BFS.
    **for** $i$ **in** $\{1..n\}$ **do**
        $atom\_probs \leftarrow \text{Softmax}(D_A(\text{SumPool}(F_A(G_{:i}))))$
        **if** $i == n$ **then**
            $loss \leftarrow \text{CrossEntropy}(atom\_probs, \text{StopToken})$
        **else**
            $loss \leftarrow \text{CrossEntropy}(atom\_probs, V_{i+1})$
            $edge\_probs \leftarrow \text{Zeros(i, b)}$
            $edge\_accum \leftarrow \text{Zeros(i, b)}$
            $edge\_accum[:, 0] \leftarrow 1$          ▷ Initialize with all unbonded edge types
            **for** $j$ **in** $\{1..i\}$ **do**
                $edge\_accum[: j] \leftarrow A_{i+1,:j-1}$
                $V'_{:i} \leftarrow \text{StackRows}(V_{:i}, V_{i+1}, edge\_accum)$   ▷ $V'_{:i} \in \mathbb{R}^{(2d+b) \times i}$. Row $j$ contains
                                                type of atom $j$, type of atom $i + 1$,
                                                  type of edge $A_{i+1,j}$
                $G'_{:i} \leftarrow (V'_{:i}, A_{:i}, X_{:i})$
                $h\_edge \leftarrow D_E(F_E(G'_{:i}))$
                $edge\_probs[j] \leftarrow \text{Softmax}(h\_edge[j])$
            **end for**
            $loss \leftarrow loss + \text{CrossEntropy}(edge\_probs, A_{i+1,:i})$
            $G'_{:i} \leftarrow (\text{StackRows}(V_{:i}, V_{i+1}, A_{i+1,:i}), A_{:i}, X_{:i})$
            $dists \leftarrow ||X_{i+1} - X_{:i}||$         ▷ $dists \in \mathbb{R}^i$. Distances to the new atom.
            $dist\_probs \leftarrow \text{Softmax}(D_D(F_D(G'_{:i})))$
            $loss \leftarrow loss + \text{CrossEntropy}(dist\_probs, \text{make\_bins}(dists))$
            $h\_angle \leftarrow F_\theta(G'_{:i})$
            $angle\_loss \leftarrow 0$
            $angle\_count \leftarrow 0$
            **for** $j$ **in** $\{1..i\}$ **do**
                **for** $k$ **in** $\{1..j-1\}$ **do**
                    **if** $\text{argmax}(A_{j,k}) > 0$ **and** $\text{argmax}(A_{i+1,k}) > 0$ **then**
                        $angle\_probs \leftarrow \text{Softmax}(D_\theta(\text{Concat}(h\_angle[j], h\_angle[k])))$
                        $angle \leftarrow \text{Angle}(X_{i+1} - X_k, X_j - X_k)$
                        $angle\_loss \leftarrow angle\_loss + \text{CrossEntropy}(angle\_probs, \text{make\_bins}(angle))$
                        $angle\_count \leftarrow angle\_count + 1$
                    **end if**
                **end for**
            **end for**
            $loss \leftarrow loss + angle\_loss/angle\_count$
        **end if**
        $\phi \leftarrow \phi - \eta \nabla_\phi loss$
    **end for**
**end while**

---

# G  ABLATION STUDIES

In order to determine which features of GEN3D allow for it to exceed the chemical and geometric accuracy of previous models like G-SchNet and E-NF, we performed a systematic ablation study of various GEN3D subnetworks. We hypothesized that the ability of GEN3D to generate and analyze chemical bonding information (rather than exclusively generating and analyzing atom types and positions) substantially contributes to its ability to generate realistic molecules. In order to ablate this feature, we replaced GEN3D's edge network with a neural network that selects a single previously generated atom as a "focus atom," which will be nearby the next atom added to the molecular graph. We also deleted all edge labels from the input data, leaving only the atom types and positions for analysis. This setup is very similar to G-SchNet. We found that this model (which we call "GEN3D Focus Atom") performed similarly to G-SchNet, and generated molecules with the correct valence 82% of the time after training on QM9. Allowing the network to generate and analyze bonding information significantly improved performance. Adding angle predictions and enforcing valence constrains during the generative process further improved chemical and geometric realism. It should be noted that, while both of these features improve performance, they are difficult to implement in models like G-SchNet and GEN3D Focus Atom, neither of which generate bonding information.

Because G-SchNet and GEN3D Focus Atom do not generate bonding information, a secondary tool must be used to infer the bonds between atoms and assess whether or not atoms have the correct valences. To accomplish this, we used the same OpenBabel-based procedure contained in the G-SchNet reference implementation. In order to guarantee a fair comparison, we also applied this procedure to molecules for which GEN3D generated its own edges, in order to eliminate the possibility that the validity gap between GEN3D and G-SchNet is caused by errors in OpenBabel's determination of bonding information. Even when inferring bonds using OpenBabel, the GEN3D models that generated edges achieved much higher validity rates that G-SchNet and GEN3D Focus Atom. The results of the ablation studies are shown in Table 4.

Table 4: Performance of Ablated Models on QM9 Molecules (3D models with hydrogens)

| Model | Atom Stability | Mol Stability | Distance JS |
|---|---|---|---|
| G-SchNet | 98.7% | 77.0% | .0031 |
| GEN3D Focus Atom | 98.7% | 82.0% | .0037 |
| GEN3D No Angles(w/o check, OpenBabel edges) | 99.4% | 93.4% | .0030 |
| GEN3D No Angles (w/o check, GEN3D edges) | 99.6% | 95.7% | .0030 |
| GEN3D No Angles (w/ check, OpenBabel edges) | 99.5% | 95.5% | .0030 |
| GEN3D No Angles (w/ check, GEN3D edges) | 99.7% | 98.0% | .0030 |
| GEN3D (w/o check, OpenBabel edges) | 99.6% | 96.4% | .0014 |
| GEN3D (w/o check, GEN3D edges) | 99.7% | 97.5% | .0014 |
| GEN3D (w/ check, OpenBabel edges) | 99.75% | 97.6% | .0014 |
| GEN3D (w/ check, GEN3D edges) | 99.87% | 99.1% | .0014 |
| QM9 (truth) | 99.99% | 99.9% | 0 |

# H   GENERATION PROCEDURE

GEN3D is an autoregressive model that augments a partially completed molecular graph. We denote a partially completed graph with $i$ atoms as $G_{:i} = (V_{:i}, A_{:i}, X_{:i})$. $V_{:i} \in \mathbb{R}^{i \times d}$ is a list of one-hot encoded atom types (i.e., the different chemical elements appearing in the dataset), and $d$ is the number of possible atom types. $A_{:i} \in \mathbb{R}^{i \times i \times b}$ is an adjacency matrix recording the one-hot encoded bond type between each pair of atoms, with $b$ representing the number of bond types. $X_{:i} \in \mathbb{R}^{i \times 3}$ is a list of atom positions. For the adjacency matrix $A_{:i}$, we include an extra bond type indicating that the atoms are not chemically bonded (unbonded atoms are still connected in the sense that information can propagate between them during the EGNN computation).

The addition of a new atom proceeds in three steps. First, a new atom type is selected as follows:

$$H = \text{SumPool}(F_A(G_{:i}))$$

$$V_{i+1} \sim \text{Categorical}(\text{Softmax}(D_A(H))),$$

where $V_{i+1}$ is the one-hot encoded type of the new atom, and $D_A$ is a neural network that decodes the EGNN graph embedding into a set of softmax probabilities. We implement $D_A$ as a 3-layer MLP. Note that, in addition to all of the atom species in the training set, we allow $V_{i+1}$ to take on an extra "stop token" value. If this value is generated, the molecule is complete, and generation terminates.

The next step in the generation procedure is to connect the new atom to the existing graph with edges. We do this in a similar manner to GraphAF, and query every atom sequentially to determine its new bond type, updating the adjacency list as needed (Shi et al., 2020). More formally, this procedure works as follows:

- Initialize $E \in \mathbb{R}^{i \times b}$ as a matrix containing each atom's edge type to the new atom $V_{i+1}$. At initialization, let $E$ contain all unbonded edge types.
- for $j$ in $1..i$ do:
    - $V'_{:i} = \text{Concat}(V_{:i}, E, V_{i+1})$ is a $i \times (2d + b)$ matrix of modified atom features. Row $j$ contains the one-hot encoded type of atom $j$, the one-hot encoded type of atom $j$'s current edge to atom $i + 1$, and the one-hot encoded type of atom $i + 1$.
    - $G'_{:i} = (V'_{:i}, A_{:i}, X_{:i})$
    - $h_j = F_E(G'_{:i})_j$
    - $A_{i+1,j} \sim \text{Categorical}(\text{Softmax}(D_E(h_j)))$. $A_{i+1,j}$ is a sampled bond type between atom $j$ and atom $i + 1$. $D_E$ is another MLP decoder that acts on the node-specific embedding of atom $j$.
    - $E_j \leftarrow A_{i+1,j}$

Through this procedure, a set of bonds is sampled for the new atom.

In the final step, the new atom is given a 3D position. We accomplish this by predicting a discrete distribution of distances from each atom in the graph to the new atom, and a discrete distribution of bond angles between edges that contain the new atom and all adjacent edges. These predictions induce a distribution over 3D coordinates. We compute this distribution by calculating how much each point in 3D space would violate the distance and angle constraints generated by the model, and giving higher probability density to points that violate the constraints less. In a secondary step, we approximately sample from this spatial distribution by drawing points from a fine, stochastic 3D grid using the likelihood function given by the distance and angle predictions. More formally, the positions of the atoms are predicted as follows:

$$V'_{:i} = \text{StackRows}(V_{:i}, E, V_{i+1})$$

$$G'_{:i} = (V'_{:i}, A_{:i}, X_{:i})$$

$$h_1, .., h_i = F_D(G'_{:i})$$

$$h'_1, .., h'_i = F_\theta(G'_{:i})$$

$$p_j = \text{Softmax}(D_D(h_j)) \text{ for } j = 1..i$$

$$q_{jk} = \text{Softmax}(D_\theta(h'_j, h'_k)) \text{ for } j, k \text{ s.t. } argmax(A_{i+1,k}) > 0 \text{ and } argmax(A_{j,k}) > 0,$$

where $D_D$ and $D_\theta$ are MLP decoders as before. Note that we are reusing the matrix $E$ from the edge prediction step, which has accumulated all of the new edges to atom $i + 1$. The probability vectors $p_1, ..., p_i$ now define discrete distributions over the distances between each atom in the graph and the new atom, and the vectors $q_{jk}$ define distributions over bond angles. If we treat these distributions as being independent, we can use the product rule to compute the likelihood of any point in 3D space:

$$L_D(X_{i+1}) = \prod_{j=1}^{i} p_j(||X_{n+1} - X_j||)$$

$$L_\theta(X_{i+1}) = \prod_{(j,k)\in I} q_{jk}(\text{Angle}(X_{i+1} - X_k, X_j - X_k))$$

$$L(X_{i+1}) = L_D(X_{i+1}) \times L_\theta(X_{i+1}),$$

where $I$ is the set of incident edges to the neighbors of atom $i + 1$, and "Angle" denotes the angle between two vectors. To sample a point from the likelihood $L(X_{i+1})$, we simply assign a likelihood to every point in a fine, stochastic grid surrounding the atoms that are bonded to atom $i + 1$, and sample from it as a categorical distribution to produce a new spatial location.

By repeating this procedure until termination, GEN3D produces a 3D molecule from a single starting atom. Note that, because the generation process is sequential, it is simple to mask out atom or edge selections that would violate valence constraints, thereby guaranteeing that generated molecules follow basic chemical rules. It is also possible for the model to predict a non-terminating atom, but then predict that no edges connect to that atom. In this case, we rerun the edge sampling procedure until at least one edge to the new atom is generated. If no edge to the new atom is produced after 10 resampling attempts, the new atom is discarded and the generation process is said to have terminated.

