# OpenReview forum: "Generating Realistic 3D Molecules with an Equivariant Conditional Likelihood Model"
_ICLR.cc/2022/Conference — ICLR 2022 Submitted_

### Official Review · Reviewer_Qzyh · 2021-10-31

**Correctness:** 3
**Technical Novelty And Significance:** 2
**Empirical Novelty And Significance:** 2
**Recommendation:** 5
**Confidence:** 4

**Main Review:**

**Strengths**
* The proposed model is a nice extension of previous autoregressive graph generative models, e.g., GCPN. Inspired by the mechanism of distance geometry and internal coordinates (e.g., pairwise distance and angles), the authors propose to autoregressively predict the next atom/edge type, next atom distances/angles, thus enabling the model to concurrently generate molecular graphs and 3D geometries. The whole methodology is classic and easy to follow.
* They experimentally validate the effectiveness of the proposed model. The experiment design seems good as it covers several research topics ranging from 2D molecular graph generation, 3D molecule design, to molecular conformation generation.
* They plan to release the code in the future.

**Weaknesses**
* The novelty of the work is limited. Compared to the previous autoregressive graph generative models (e.g., GCPN, GraphAF, MolecularRNN[1,2,3]), the key difference is that they incorporate the atomic distance prediction and angle prediction into their model. However, distance geometry modeling and angle prediction are not something new. So the model is more like a combination of techniques in 2D graph generation and 3D structure modeling.
* Since the paper introduces a **new generative model** for 3D molecule design, I recommend authors introduce the generative model in a more formal (mathematical) way. For example, we can introduce the probabilistic distribution we want to model, the way we factorize the distribution (autoregressively), and how we estimate the parameters. This is a minor suggestion, but I think it will improve the quality of the draft.

**Reference**
1. Graph Convolutional Policy Network for Goal-Directed Molecular Graph Generation
2. GraphAF: a Flow-based Autoregressive Model for Molecular Graph Generation
3. MolecularRNN: Generating realistic molecular graphs with optimized properties

Post rebuttal: The authors partially addressed my concerns and I would like to update the score to 5

**Summary Of The Paper:**

* The paper proposes an autoregressive generative model for 3D molecular graph generation, by predicting the next atom type, next edge type, next atom distances, and next atom angles sequentially based on the existing graph.
* The authors claim that the proposed model can generate 3D molecules with higher rates of chemical validity, better atom-distance distributions, and is very cheap to train compared with previous models.

**Summary Of The Review:**

My main concerns of the paper are its novelty and the way it introduces the generative model.

---

> ### Author Response · Authors · 2021-11-17
> **Response to Reviewer Qzyh**
>
> We thank the reviewer for taking the time to review our paper and provide this useful feedback.  We have tried to thoroughly address all of the comments in the following response.
>
> >The novelty of the work is limited. Compared to the previous autoregressive graph generative models (e.g., GCPN, GraphAF, MolecularRNN[1,2,3]), the key difference is that they incorporate the atomic distance prediction and angle prediction into their model. However, distance geometry modeling and angle prediction are not something new. So the model is more like a combination of techniques in 2D graph generation and 3D structure modeling.
>
> We agree with the reviewer that our model can be accurately characterized as a combination of techniques in 2D graph generation and 3D molecular modelling.  We feel, however, that the novel synthesis of two existing, previously independent directions of research into a coherent mathematical and computational framework makes this contribution foundational for future application-oriented studies.  The joint generation of molecular graphs and coordinates allows us to optimize generated molecules for properties that require both 3D coordinates and molecular graph information; we have now modified the text to emphasize our demonstration of this in Appendix E.  Furthermore, GEN3D achieves state-of-the-art results in multiple well-studied tasks in molecular machine learning, including the generation of realistic molecular graphs, the generation of realistic atomic coordinates, and the prediction of molecular geometries, underscoring the relevance of our contribution.  Our new ablation studies confirm that the joint generation of molecular graphs and 3D coordinates is crucial for achieving these state-of-the-art results.  Finally, we believe the combination of graph and geometric generation demonstrated by GEN3D could have implications for 3D-graph generation tasks in many domains aside from molecular design, making our method an important advance.
>
> >Since the paper introduces a new generative model for 3D molecule design, I recommend authors introduce the generative model in a more formal (mathematical) way.
>
> We thank the reviewer for this valuable suggestion.  We have updated our manuscript with a more formal characterization of our generative model in terms of the target distribution, factorization, and method of estimation.  We found that this characterization makes the text more clear and concise.  The probabilistic distribution that underlies GEN3D suggests possible extensions of the model by conditioning on additional contextual information, such as surrounding protein atoms.

---

### Official Review · Reviewer_7FsH · 2021-11-02

**Correctness:** 3
**Technical Novelty And Significance:** 2
**Empirical Novelty And Significance:** 2
**Recommendation:** 5
**Confidence:** 3

**Main Review:**

- Pros
    - Highlights the importance of generating 3d structure for molecule generation.
    - The paper provides new insight of the QM9 dataset.

- Cons
    - Insufficient description. The paper seems not-self contained. Difficult to understand some of the details.
        - Since ICLR is not chemistry focused conference, it would be good to provide some details of domain-specific information
    - The 3D structure generation is unclear.
        - To generate 3D coordinates, the model places a probability mass over the grids of distances and angles.
        - The provided process is likely to place a mass over the impossible configuration of distances and angles.
        - From the grid, the model sample a new location. However, sampling is not back-propagatable in general. It is unclear how the model handles this.
    - A mismatch between motivation and experiments
        - None of the experiments is related to the proposed downstream tasks in the introduction. Although generating 3D structure seems plausible, the benefit of it is unclear without direct application of 3D structure
            - JS divergence on distance is the only metric that is directly connected to the molecule geometry. However, it is not sufficient to show the usefulness of 3D structure.
        - For example, it would be more interesting to show that the generated graphs have relatively low-energy conformations than those from the other models.
    - A proper ablation study on model selection and training procedures would improve the strength of the proposed approach.

- Questions
    - What are the losses used to train the model? Please provide the detailed training procedure.
    - Is the equivariance of the subnetworks sufficient to impose the equivariance of the entire network?
        - Is permutation invariant unnecessary? Since the generation is autoregressive, to put the same distribution to the same molecule with different sequences, it needs to be permutation invariant.
    - If the description of QM9 dataset is correct, how do we understand the reported performance of some previous models which achieve 90%+ uniqueness and novelty such as GraphDF[1] and MoFlow[2]. This seems impossible based on the description provided in this paper.
    - In the Introduction, a significant portion has been used to criticize the approaches with SMILES. What problem can be arisen from converting SMILES string to 3D structure? and how the proposed solution can avoid this? - e.g., w. OpenBabel
    - What is the GEOM-QM9 dataset, and how does it differ from QM9?
        - In the last sentence of Section 3, a random uniform noise is added to each coordinate. Is this applied to all datasets?

1. Luo, Youzhi, Keqiang Yan, and Shuiwang Ji. "GraphDF: A discrete flow model for molecular graph generation." arXiv preprint arXiv:2102.01189 (2021).
1. Liu, Meng, et al. "GraphEBM: Molecular graph generation with energy-based models." arXiv preprint arXiv:2102.00546 (2021).

**Summary Of The Paper:**

This paper proposes a generative model to sample a new molecule with its 3d coordinates. The generative process is divided into three different steps: 1) the atom network generates a new atom 2) the edge network connects the new atom to the existing atoms 3) the distance and angle network generates the pairwise distances and angles from existing atoms to the new atoms. EGNN is used for each of these sub-networks to impose the equivariance to translations and rotations. Experiments with QM9 datasets show the geometric correctness of the proposed generative process.


**Summary Of The Review:**

Although the proposed method seems an important research direction for molecule generation, the experiments did not show the usefulness of such approaches to compare with the previously studied methods. Some of the descriptions are insufficient to understand the model details as well.

---

> ### Author Response · Authors · 2021-11-17
> **Response to Reviewer 7FsH (3/3)**
>
> >**Questions**
>
> >What are the losses used to train the model? Please provide the detailed training procedure.
>
> We use cross-entropy losses.  In the language model example above, a cross-entropy loss can be used to measure the similarity between the predicted discrete distribution of tokens and the actual next token in the sentence.  Similarly, our model predicts discrete distributions of atom types, edges, distances, and angles, which can also be optimized using a cross-entropy loss.  We have provided the detailed training procedure in a new Appendix.
>
> >Is the equivariance of the subnetworks sufficient to impose the equivariance of the entire network?
>
> We have updated the manuscript to better explain why it is sufficient.  The density of our generative model can be factored into a product of multiple sub-distributions that dictate the density of each new atom, edge, and location in the molecule.  Because each of these individual densities is insensitive to rotations and translations, the product density is also insensitive to these transformations.  As a consequence, if an arbitrary rotation or translation is applied during the middle of the model’s generative process, the end result will be the same 3D molecule with that corresponding rotation and translation.
>
> >Is permutation invariant unnecessary? Since the generation is autoregressive, to put the same distribution to the same molecule with different sequences, it needs to be permutation invariant.
>
> Like most previous models that generate molecules, our model’s density is not invariant to the sequence in which the atoms of a molecule are presented.  We thus randomize the sequence in which atoms are presented at training time, so the model should learn to ascribe similar densities to different rollouts of the same molecule.
>
> >If the description of QM9 dataset is correct, how do we understand the reported performance of some previous models which achieve 90%+ uniqueness and novelty such as GraphDF[1] and MoFlow[2]. This seems impossible based on the description provided in this paper.
>
> Similar to what we found with CGVAE and GraphAF, it is likely that much of this novelty comes from molecules with more than nine heavy atoms.  The QM9 dataset is exhaustive because it is a subset of molecules, with up to nine heavy atoms, from the GDB-17 database, which exhaustively enumerates all molecular graphs with up to 17 heavy atoms.  In Figure S2, we document the tendency of other models to produce molecules with more than nine heavy atoms, and to produce molecular graphs that are geometrically unviable.
>
> >In the Introduction, a significant portion has been used to criticize the approaches with SMILES. What problem can be arisen from converting SMILES string to 3D structure? and how the proposed solution can avoid this? - e.g., w. OpenBabel
>
> While it is possible to convert from generated SMILES strings to a 3D structure using standalone methods, this does not allow for the geometry of the molecule to be learned and optimized by the model.  For example, we show in Appendix E that GEN3D can learn to produce molecules in conformations that are conducive to binding a protein of interest.  Tools that generically convert molecular graphs to 3D coordinates are not able to fine-tune these coordinates for specific contexts and applications.
>
> >What is the GEOM-QM9 dataset, and how does it differ from QM9?
>
> GEOM-QM9 is a dataset recently introduced by Gómez-Bombarelli and Axlerod (https://arxiv.org/abs/2006.05531).  The dataset contains the same set of compounds as QM9, but each compound is associated with multiple conformations that represent different energy minima in the quantum mechanical energy function.  This dataset has become a standard benchmark for methods that predict molecular geometries from molecular graphs.
>
> >In the last sentence of Section 3, a random uniform noise is added to each coordinate. Is this applied to all datasets?
>
> Yes, the noise is applied to all datasets.  We have updated the manuscript to make this more clear.

---

> ### Author Response · Authors · 2021-11-17
> **Response to Reviewer 7FsH (2/3)**
>
> >None of the experiments is related to the proposed downstream tasks in the introduction. Although generating 3D structure seems plausible, the benefit of it is unclear without direct application of 3D structure
>
> We agree with the reviewer that there are many possible applications for GEN3D that we have not explored.  We have now modified the text to emphasize a property optimization task in Appendix E, however, in which we used GEN3D to generate molecules that have molecular graphs and geometries that are predicted to bind in a protein pocket.  Our results show that both the molecular graphs and 3D coordinates constructed by GEN3D contribute to our ability to generate successful binders, thereby underscoring the promise of our method in drug discovery.
>
> >JS divergence on distance is the only metric that is directly connected to the molecule geometry. However, it is not sufficient to show the usefulness of 3D structure. For example, it would be more interesting to show that the generated graphs have relatively low-energy conformations than those from the other models.
>
> Our exploration of property optimization in Appendix E demonstrates the utility of 3D structures in downstream applications.  More importantly than the JS divergence metric, we have established the geometric accuracy of our model by predicting molecular geometries conditionally on specific molecular graphs—a task that has been investigated in a number of prior publications—and achieved this with state-of-the-art results.  The fact that our model does a better job than previous approaches at generating conformations that match those in the GEOM-QM9 dataset on arbitrary unseen molecules is an indication that our model has a strong bias towards stable, low-energy conformations, as the poses in GEOM-QM9 were created through the minimization of the quantum mechanical energy.  See our response below for a more detailed description of GEOM-Q9.
>
> While it would be interesting to calculate the quantum mechanical energy of the molecules we have generated, it would be difficult to make direct comparisons with other generative models, as the models would not produce the same set of molecular graphs.  When comparing the quantum mechanical energies of distinct molecules, it is important that they correspond to the same molecular graph, as different molecular graphs can have vastly different baseline energies.
>
> >A proper ablation study on model selection and training procedures would improve the strength of the proposed approach.
>
> We agree with the reviewer that ablation studies are important for explaining the empirical success of machine learning techniques.  In response to this suggestion, we have conducted a systematic ablation study in which we successively removed various elements of the GEN3D architecture.  We found that both the chemical and geometric accuracy of generated molecules declined continuously as we ablated GEN3D’s angle prediction network and removed its ability to generate chemical bonding information.  Removing GEN3D’s ability to generate chemical bonding information resulted in a baseline model that solely generated atomic coordinates—similar to previously published models like G-SchNet and ENF.  This baseline model performed similarly to G-SchNet, confirming that GEN3D’s methodologically novel ability to generate both bonding information and atomic coordinates is crucial for its state-of-the-art performance.

---

> ### Author Response · Authors · 2021-11-17
> **Response to Reviewer 7FsH (1/3)**
>
> We thank the reviewer for taking the time to review our paper and  provide this useful feedback.  We have tried to thoroughly address all of their comments in the following response.
>
> >Since ICLR is not chemistry focused conference, it would be good to provide some details of domain-specific information
>
> Thank you for this suggestion. We have added two additional paragraphs to the Introduction of the paper that discuss the basic principles of atomic bonding and the molecular geometry of organic molecules.
>
> >The provided process is likely to place a mass over the impossible configuration of distances and angles.
>
> It is true that, if we were to sample from the distribution of distances and angles specified by the model predictions, these constraints would almost certainly be impossible to satisfy.  To resolve this issue, we add the log-likelihood contributed by each distance and angle prediction to a set of points in 3D space.  Each point in 3D space can be thought of as accumulating a “loss” based on how much it would violate all of the angle and distance constraints generated by the model.  No point will perfectly satisfy the constraints, but some will have a better loss than others.  We then sample points such that those with lower loss are more likely to be sampled, and we have found that this works well in practice.  We have updated the manuscript to make this more clear.
>
> >From the grid, the model sample a new location. However, sampling is not back-propagatable in general. It is unclear how the model handles this.
>
> Because the model is autoregressive, there is no need to back-propagate through the sampling process.  By way of comparison, autoregressive language models may predict a discrete probability distribution over the next token in a sentence.  The cross-entropy loss between this predicted distribution and the true next token can be used to update the model weights during training.  Sampling from the predicted distribution only happens when sampling a new sentence after the model has been trained.  We have updated the manuscript to make this more clear.

---

### Official Review · Reviewer_biHr · 2021-11-02

**Correctness:** 3
**Technical Novelty And Significance:** 1
**Empirical Novelty And Significance:** 2
**Recommendation:** 3
**Confidence:** 4

**Main Review:**

1. The related work is pretty detailed but neglects some relevant related papers.
   1) Simm, G. N. C.; Pinsler, R.; Hernández-Lobato, J. M. Reinforcement Learning for Molecular Design Guided by Quantum Mechanics. ICML 2020.
   1) Gebauer, N. W. A.; Gastegger, M.; Hessmann, S. S. P.; Müller, K.-R.; Schütt, K. T. Inverse Design of 3d Molecular Structures with Conditional Generative Neural Networks. arXiv:2109.04824 [physics, stat] 2021.
   1) Simm, G. N. C.; Pinsler, R.; Csányi, G.; Hernández-Lobato, J. M. Symmetry-Aware Actor-Critic for 3D Molecular Design. ICLR 2020.

2. It is not clear how working with a graph is beneficial. 1) The authors claim that organic chemistry rules can easily be enforced through a graph representation. However, it's unclear to what extent these relatively simple rules are holding ML models back from proposing more chemically meaningful molecules. 2) Working with a graph restricts the application of this model to organic chemistry, and metals or multi-molecular clusters become unattainable.

**Summary Of The Paper:**

The authors propose GEN3D, an ML-based approach for 3D molecular structure generation. GEN3D places atoms in space to build molecules in an atom-by-atom fashion and constructs a corresponding molecular graph that can be employed to enforce organic chemistry rules.

**Summary Of The Review:**

GEN3D appears to be a competitive approach for molecular structure generation. The main weakness of this paper is, however, that it lacks methodological novelty. Methods, such as G-Schnet, MolGym, and ENF, are closely related to this work, and the originality of this work is, in my view, insufficient for this venue.

---

> ### Author Response · Authors · 2021-11-17
> **Response to Reviewer biHr**
>
> We thank the reviewer for taking the time to review our paper and provide this useful feedback.  We have tried to thoroughly address all of their comments in the following response.
>
> >The related work is pretty detailed but neglects some relevant related papers.
>
> We thank the reviewer for pointing out these relevant papers.  We have updated the manuscript to cite these works, and describe how they relate to GEN3D.
>
> >It is not clear how working with a graph is beneficial...
>
> In principle all of the essential information for describing a molecule is present in the 3D configuration of its atoms, without the need for explicitly considering chemical bonds.  Our paper, however, demonstrates that training on and generating molecular graphs, in addition to coordinates, substantially improves performance on a variety of applications, and we have emphasized this point in our revised manuscript.  In comparison to G-SchNet, which generates coordinates without the corresponding bonds, GEN3D generates 25 times fewer molecules with incorrect valences, and this difference in failure rate will grow exponentially when generating larger molecules.  GEN3D also dramatically outperforms the accuracy of a very recently published extension of G-SchNet called G-SphereNet (https://openreview.net/forum?id=C03Ajc-NS5W); the failure rate of GEN3D is about 13 times smaller than that of G-SphereNet when the models are trained on the same dataset (comparing Table 2 of our paper with Table 1 of G-SphereNet), and this manifests itself in a higher quality for our generated 3D molecules.
>
> The new ablation studies that we will include in the revised manuscript confirm that this gain in accuracy is due to GEN3D’s ability to generate and analyze bonding information in addition to atomic coordinates, illustrating the benefit of working with molecular graphs in addition to coordinates.  Our new ablation studies also found that adding angle predictions and enforcing valence constraints during the generative process further improved chemical and geometric realism, but these performance-enhancing features would be difficult to implement in models like G-SchNet that  do not generate bonding information.
>
> Generating molecular graphs is also important because many downstream applications require them, including chemical synthesis, molecular dynamics simulation, and virtual screening.  We have modified the text to emphasize such an application in our docking study in Appendix E.  Though it is often possible to obtain graphs from the geometries produced by models like G-SchNet using tools like OpenBabel, this introduces a non-learnable and potentially error-prone step that could negatively impact the quality of the generated molecules.  Because it directly generates molecular graphs, GEN3D suffers from no such drawbacks.  Finally, the joint generation of molecular graphs and geometries allows for interesting applications like generating molecular conformations conditional on a fixed molecular graph, which we demonstrate in this paper with state-of-the-art results.  We feel that this is the most compelling demonstration to date of geometric accuracy in a generative model for 3D molecules.
>
> >Working with a graph restricts the application of this model to organic chemistry, and metals or multi-molecular clusters become unattainable.
>
> Our framework allows for non-bonded edges to be present in molecular structures, so it is entirely possible to represent multi-molecular clusters or metal atoms with metallic or intermolecular interactions represented as virtual edges, and intramolecular covalent interactions represented as bonded edges.  Such consideration of both bonded and non-bonded interactions is common in molecular dynamics simulation.  Furthermore, our focus is primarily in drug discovery, so organic molecules are of particular interest to us.

---

### Official Review · Reviewer_uzvM · 2021-11-03

**Correctness:** 3
**Technical Novelty And Significance:** 3
**Empirical Novelty And Significance:** 3
**Recommendation:** 6
**Confidence:** 3

**Main Review:**

Pros:
 - Problem is relevant and clearly defined
 - Method is intuitive and accessible - generate atom type, generate connections to existing atoms, generate positions, repeat
 - Some results are positive, generally better or similar to other methods. The model's tendency to construct valid molecules without checks is impressive.
 - Writing is clear, organized, and developed, discrepancies are discussed

Cons:
 - The result in Table 1 make me question the overall value of the technique.
   - The method w/o geometry has higher validity and uniqueness than the model w/ geometry.
   - All of the methods (w/ checks) produce valid molecules and have similar uniqueness and higher novelty. The paper discusses this apparent shortcoming and largely concludes that the comparison is misleading due to intentional limitations (number of atoms) in the proposed method vs the other methods. If this is true, is it possible to reproduce Table 1 where only "fair" comparisons are made (maybe by dropping large molecules from the other methods)?
 - The paper discusses how the conditional model could be used to retain features from one molecule when constructing new molecules but doesn't demonstrate this. It would strengthen the paper to see a simple demonstration of this application.

**Summary Of The Paper:**

The paper proposes a model to generate molecules using graph and geometric representations. The method is tested against several relevant baselines on standard and custom datasets, with a primary focus on chemical validity.

**Summary Of The Review:**

The paper takes a positive step forward in constructing generative models over molecules by including graphical and geometrical information in a single model. However, some of the empirical evaluations muddle the overall story and could be improved/clarified.

---

> ### Author Response · Authors · 2021-11-17
> **Response to Reviewer uzvM**
>
> We thank the reviewer for taking the time to review our paper and provide this useful feedback.  We have tried to thoroughly address all of their comments in the following response.
>
> >The result in Table 1 makes me question the overall value of the technique
>
> We believe that unchecked chemical validity is an important indicator of how well a machine learning model can learn to generate molecules from a target region of chemical space.  While it is true that previously presented models can generate valid molecules when checks are applied, a low unchecked validity rate raises concerns about how faithfully the generated molecules recapitulate other chemical properties of the target distribution, even when checks are applied.  Our goal in this work was to develop a model that samples from the joint distribution of molecular graphs and 3D atomic coordinates.  For such a model to be deemed accurate, we felt that it should (among other things) be able to match the unchecked validity rates of previous molecular graph generators.  GEN3D cleared this bar, and ultimately exceeded the validity of previous methods, which is a promising sign for further tests and applications of the method.
>
> It is true that our geometry-ablated model achieves a slightly higher unchecked validity rate than GEN3D.  This may be because the accumulation of small geometric errors in the 3D model reduces accuracy in later iterations (which is a classic problem for models that use teacher forcing).  The primary benefit of jointly sampling graphs and 3D coordinates, however, is not a reduction in unchecked errors.  The real value added is in the ability to optimize geometry-dependent molecular properties, so the slight drop in validity relative to GEN3D− does not significantly detract from the added value of the method.  In our view, Table 1 of the paper confirms that GEN3D generates molecular graphs with high chemical validity.  The fact that GEN3D and GEN3D− show significantly higher validity than previous models is somewhat surprising, especially because GEN3D− can robustly outperform other graph-generation methods, despite being conceptually similar.  The ultimate value of GEN3D in applications extends far beyond chemical validity rates, however, and additional applications are explored later in the paper.
>
> With regard to the novelty scores, we would like to emphasize that there are no limitations on the number of generated atoms in our method.  In contrast, CGVAE and GraphAF impose hard constraints on the number of atoms generated (at 10 and 38 atoms, respectively).  Empirically, GEN3D does a better job generating molecules that stay within the nine-atom limit found in the QM9 dataset, but this is a consequence of the model having learned to obey this limit, and not an intentional cap on the number of atoms.  Because GEN3D learns to stay within the limit of nine atoms so accurately, and QM9 is an exhaustive dataset, this leads to lower novelty scores, approaching the percentage of the dataset that was withheld during training—the optimal outcome for a model that aims to generate the region of chemical space enumerated by QM9.  Novelty scores reported above the proportion of the QM9 dataset held out during training indicate the frequent generation of molecules that do not conform to the rules of QM9; our analyses in Figure S2 demonstrate that this is the case for CGVAE.  Because we have not given GEN3D any advantage in our comparisons to other generators, we believe the current comparison is fair.
>
> >The paper discusses how the conditional model could be used to retain features from one molecule when constructing new molecules but doesn't demonstrate this. It would strengthen the paper to see a simple demonstration of this application.
>
> We have modified the text to emphasize the results in Appendix E, in which we show that GEN3D can be fine-tuned to generate novel molecules with both molecular graphs and molecular geometries that are conducive to binding in a specific protein pocket.  In this application, GEN3D retains both geometric and chemical information from the fine-tuning set when constructing new samples, leading to the generation of new molecules with better binding properties.

---

### Author Response · Authors · 2021-11-17
**Response to All Reviewers (2/2)**

GEN3D has also been used to generate molecules with up to 50 heavy atoms, spanning the range of most drug-like molecules, while previous 3D molecular generators have only been trained on QM9, and have not demonstrated the ability to generate valid molecules of sizes relevant to drug discovery.  The difference in validity rates is especially important in this context because the probability of generating invalid molecules increases exponentially with the size of the molecule.  In addition, the concurrent generation of molecular graphs and 3D coordinates in GEN3D allows for generated molecules to be used in applications that require bonding information without the need for an external tool to infer bonding (which may introduce unintended errors).

Finally, we would like to emphasize that GEN3D has achieved state-of-the-art results in three distinct categories of tasks, each of which have been approached by specialized models in the past: the generation of valid and realistic molecular graphs; the unconditional generation of realistic 3D atomic coordinates; and the prediction of molecular geometries for specific molecular graph inputs.

The reviewers have raised additional comments about the details and presentation of the model.  We have addressed these comments in our responses to individual reviewers.  Once again, we greatly appreciate the reviewers’ time and effort in preparing these reviews, and we have used this feedback to improve our manuscript, which we will upload shortly.

---

### Author Response · Authors · 2021-11-17
**Response to All Reviewers (1/2)**

We would like to thank all of the reviewers for taking the time to read our paper and provide useful feedback.  The two main concerns raised by the reviewers are a lack of demonstrated utility from simultaneously generating molecular graphs and 3D coordinates, and a lack of novelty in the proposed approach.

With regard to the usefulness of concurrently generating molecular graphs and atomic coordinates, we have now modified the text to emphasize the experiments in Appendix E.  In these experiments, we used GEN3D to generate molecules with molecular graphs and molecular geometries that are especially conducive to binding in a protein pocket of interest.  This task requires the optimization of a molecular property that requires both 3D coordinates and bonding information to compute, making this application specifically well suited to a model like GEN3D.  We note that Reviewer 2 was skeptical of the utility of generating a molecular graph in addition to 3D coordinates, while Reviewer 4 was unsure whether the addition of 3D coordinates was a substantial improvement over graph-only generative models.  The ability of GEN3D to optimize  generated molecules for properties like predicted binding affinity, which requires bonding and geometric information, illustrates why producing both the graph and coordinates are important contributions offered by GEN3D.

With regard to novelty, it is true that models like GraphAF and GCPN can generate molecular graphs, and models like G-SchNet and ENF can generate 3D molecular coordinates.  We have demonstrated, however, that GEN3D represents a substantial improvement upon the fundamental capabilities and empirical performance of all of these models.  When compared to previous 3D generative models like G-SchNet, GEN3D produces molecular structures with substantially higher validity rates and substantially more realistic geometries.  In particular, GEN3D produces invalid molecules at a rate that is 25 times lower than G-SchNet when the models are trained on the QM9 dataset, and decreases the divergence of the distance distributions by more than a factor of two.

Another model currently under review at ICLR 2022, G-SphereNet (https://openreview.net/forum?id=C03Ajc-NS5W), was specifically developed to improve the performance of G-SchNet, and GEN3D also outperforms this model.  Additional ablation studies we performed at the suggestion of reviewer 3 demonstrated that the methodological innovation of concurrently generating molecular graphs and 3D coordinates is responsible for this considerable improvement in empirical performance.  In the ablation studies, we successively removed elements of GEN3D to produce the baseline model that is similar to G-SchNet.  We found that performance in both geometric and chemical accuracy metrics dropped continuously as we removed features, and that our baseline model performed very similarly to G-SchNet.  We show the key results of the ablation studies in the table below, and describe them in full in a new Appendix in the manuscript.

**Table 4 from the manuscript: Performance of Ablated Models on QM9 Molecules (3D Models with Hydrogens)**

| Model                                       | Atom Stability | Mol Stability | Distance JS |
| ------------------------------------------- | -------------- | ------------- | ----------- |
| G-SchNet                                    | 98.7\%         | 77.0\%        |       .0031 |
| GEN3D Focus Atom                            | 98.7\%         | 82.0\%        |       .0037 |
| GEN3D No Angles(w/o check, OpenBabel edges) | 99.4\%         | 93.4\%        |       .0030 |
| GEN3D No Angles (w/o check, GEN3D edges)    | 99.6\%         | 95.7\%        |       .0030 |
| GEN3D No Angles (w/ check, OpenBabel edges) | 99.5\%         | 95.5\%        |       .0030 |
| GEN3D No Angles (w/ check, GEN3D edges)     | 99.7\%         | 98.0\%        |       .0030 |
| GEN3D (w/o check, OpenBabel edges)          | 99.6\%         | 96.4\%        |       .0014 |
| GEN3D (w/o check, GEN3D edges)              | 99.7\%         | 97.5\%        |       .0014 |
| GEN3D (w/ check, OpenBabel edges)           | 99.75\%        | 97.6\%        |       .0014 |
| GEN3D (w/ check, GEN3D edges)               | 99.87\%        | 99.1\%        |       .0014 |
| QM9 (truth)                                 | 99.99\%        | 99.9\%        |           0 |

---

### Author Response · Authors · 2021-11-19
**Notes on the revised manuscript**

Major changes in this revision have been marked in blue. Note that we have added additional appendices, but maintained the labelling of the original appendices in order to remain consistent with the original draft and reviews. The Appendices thus appear out of order in the revised text. Should our paper be accepted, we will resolve this issue for the camera-ready version.

---

### Decision · Program_Chairs · 2022-01-20

**Decision:**

Reject

**Comment:**

This paper proposes to generate 3D molecules using a step by step approach. The reviewers raised major concerns on the experiments, novelty, writing and technical details. The authors also were not aware of many of the important references, part (but not all) of which have been included during discussions. It is clear that this work is not ready to be accepted by ICLR.